# Investigation of the Microstructure and Properties of Aluminum–Copper Compounds Fabricated by the High-Pressure Die Casting Process

**Nane Nolte [1], Thomas Lukasczyk [1,\*] and Bernd Mayer [1,2]**

[1]    Fraunhofer Institute for Manufacturing Technology and Advanced Materials IFAM, Wiener Str. 12, 28359 Bremen, Germany

[2]    Faculty of Production Engineering, University of Bremen, 28359 Bremen, Germany

\*    Correspondence: thomas.lukasczyk@ifam.fraunhofer.de

**Abstract:** The material combination of aluminum and copper is increasingly coming into focus, especially for electrical contact applications. Investigations of different casting processes show that a significant influence for the formation of a material bond is the thermal impact. For high-pressure die casting (HPDC) processes, the impact is quite low, e.g., due to short cycle times. Despite the high efficiency of this technology, currently there are hardly any investigations in this respect. So, the technology was used in this study to produce aluminum–copper compounds and analyze interfacial layers by means of SEM images and EDX measurements. Furthermore, the mechanical and electrical properties of the compounds were determined by means of tensile shear tests and measurements of the electrical conductivity. By modifying specimen geometry, the thermal impact could be increased and, thus, enhanced compound properties were achieved. Overall, compounds of sufficiently high mechanical strength, as well as electrical conductivity, could be produced by HPDC processes, demonstrating the high technical and economic potential of this casting technique.

**Keywords:** compound casting; intermetallic; composite casting; aluminum; copper; electrical conductivity; microstructure; high-pressure die casting; HPDC

## 1. Introduction

Due to its high conductivity, copper is of particular interest for various applications, e.g., in electrical automotive components or battery technology. However, the high specific weight and costs are major disadvantages of copper from an economic point of view. In order to reduce these drawbacks, copper is often replaced by lightweight and cost-effective aluminum. The lower electrical conductivity compared to copper is compensated for by an increase in the volume of aluminum, resulting in an increased assembly space for the corresponding components. Since copper cannot be entirely substituted, the need for suitable joining technologies for aluminum–copper compounds is great. In particular, the high reactivity of the metals with each other is a special challenge.

Compound casting is a very innovative and efficient approach. By integrating the joining process into the actual forming process, it is possible not only to reduce production time and costs, but also to increase complexity of the components and integrate additional functions, e.g., via adding cooling structures or sensor technology. Various studies have been conducted on compound casting of aluminum and copper using different casting methods such as continuous casting, simultaneous casting, low-pressure casting and permanent mold casting [1–4]. In the respective approaches, joining between copper and aluminum always takes place via a material bond as intermetallic phases. Depending on the thermal impact, these differ in their formation type and thickness. Li et al. [1] showed this relationship in their investigations by means of annealing experiments. Thereby, a formation and growth of the intermetallic phases dependent on the annealing time, as

well as temperature, could be detected. The thermal conditions in the casting process and, sequentially, the formation of intermetallic phases can be modified via several process parameters depending on the respective process. For example, these can be the pouring temperature [2,3], preheating temperature of the insert [4] or mold temperature/cladding temperature [5].

The disadvantages of an intermetallic phase formation between copper and aluminum are very hard and brittle layers. They are often susceptible to cracking and have significantly lower electrical conductivity than their base materials [1,4,6–8]. If the thickness of the intermetallic phase surpasses a certain value, these brittle material properties will dominate, resulting in a decreasing compound strength. This relation between decreasing compound strength and growing phase thickness has already been discussed in previous publications [1–4]. Abbasi et al. [6] described a change in fracture behavior from ductile to brittle after the thickness of the intermetallic phases reached 2.5 µm. Thus, for an ideal aluminum–copper compound, it would be necessary to form a material bond via intermetallic phases thick enough to be mechanically stable and sufficiently thin to still have high electrical conductivity.

The previously mentioned studies refer to casting processes where high thermal impact is given or can be further increased by process parameters, which result in a continuous intermetallic phase formation. In contrast to the other presented casting techniques, HPDC processes are characterized by low mold temperatures, component thicknesses and high cooling rates, resulting in more difficult conditions for an intermetallic phase formation. Yet, due to the short cycle times and the high level of automation, HPDC is particularly efficient and, thus, very attractive from an industrial point of view.

Freytag [9] reports, in his studies which produce compounds of high thermal conductivity by HPDC, that no material bond is formed for cast-on aluminum–copper specimens. A compound can only be achieved by coating the inserts. This means an additional step in production, which causes time and costs, but can also result in an increase of electrical resistance depending on the type of coating. At present, it is questionable whether it is possible to achieve a stable compound between aluminum and copper with an electrical conductivity high enough for industrial applications using the HPDC process.

Therefore, a new approach for the generation of an aluminum–copper compound in HPDC is investigated in this work. By changing the compound geometry, the thermal impact in the interfacial layer of a copper inlet can be modified. This leads to better control of the intermetallic phase formation. The resulting mechanical and electrical properties were investigated and will be discussed in relation to interface conditions.

## 2. Materials and Methods

### 2.1. Materials and Setup

To ensure the highest possible electrical conductivity of the components, pure copper inserts made of Cu-ETP (Bikar Metalle, Bad Berleburg, Germany) and rotor aluminum Al99,7 (Rheinfelden Aluminium, Baden, Germany) as the die-casting alloy were used. The chemical composition of the components is listed in Table 1.

**Table 1.** Chemical composition of the compound materials used in the experiments.

| Material | Al | Cu | Si | Fe | Mn | Mg | Bi | Zn | O | Ti | Pb | Sb |
|---|---|---|---|---|---|---|---|---|---|---|---|---|
| Al99,7-E | 99.640 | 0.002 | 0.134 | 0.181 | 0.004 | 0.001 | 0.007 | 0.015 | - | 0.001 | 0.003 | 0.005 |
| (Cu-ETP)/CW004A | - | 99.900 | - | - | - | - | - | - | 0.040 | - | 0.005 | - |

The experiments were carried out using a cold-chamber die casting system of type SC N/66 from Bühler (Uzwil, Switzerland) with a closing force of 660 t. Hybrid casting process steps are schematically depicted in Figure 1.

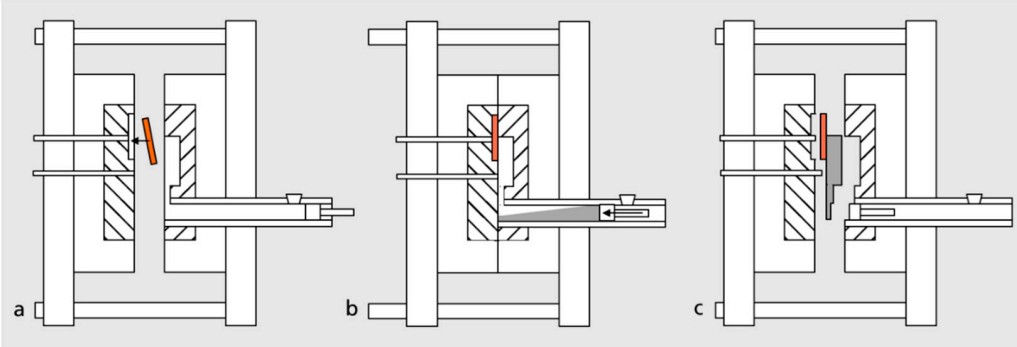

**Figure 1.** Schematic representation of the joining process of a cast-on copper insert and molten aluminum in HPDC with (**a**) mold preparation, (**b**) piston acceleration and(**c**) shakeout.

The copper inserts (40 mm × 100 mm × 1.5 mm) were placed manually in the moving die half (Figure 1a). Afterwards, the mold was closed at 200 °C so that the copper inserts were fixed in position by the two die halves. Only a few seconds after placement, the copper inserts heat up to the mold temperature. After degassing, the aluminum melt was automatically filled into the casting sleeve at a temperature of 740 °C and injected into the cavity by accelerating the casting piston in milliseconds (Figure 1b). The applied casting mold allowed for the creation of two different compound geometries, so that the copper inserts can be cast-on to an area of 40 mm × 40 mm (Figure 2a) as well as cast-in 40 mm deep (Figure 2b).

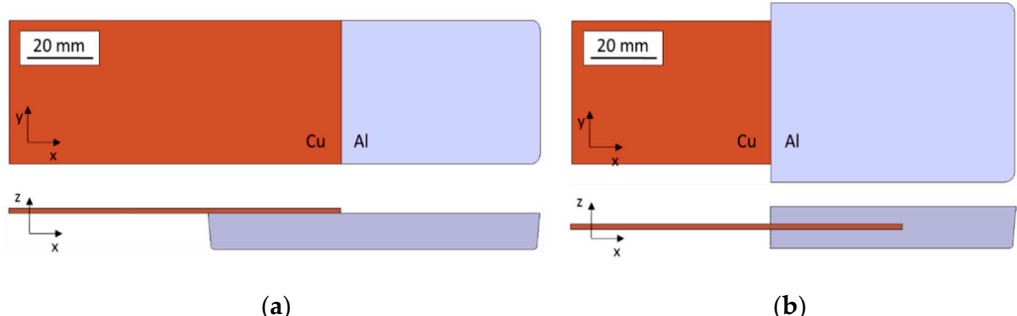

| (**a**) | (**b**) |

**Figure 2.** Schematic illustration of the applied compound casting geometry in top view and sectional view of (**a**) cast-on sample and (**b**) cast-in sample.

After manual removal from the mold (Figure 1c), the casted compounds were allowed to cool down to room temperature. Subsequently, the cast-on and cast-in specimens were mechanically cut off from the sprue and prepared for examination of the mechanical and electrical properties.

### 2.2. Analysis of the Aluminium Copper Interfacial Layer

The bond strength of the cast-on and cast-in specimens was determined in a tensile shear test on the basis of DIN EN 1465, as shown in Figure 3a. A Zwick/Roell test rig of type Z020 with a maximum load of 20 kN was used. Furthermore, micrographs were taken of the compound samples to investigate the formed aluminum–copper interface. They show the yz-plane (see Figure 2) in each case and were taken at several positions along the x-axis to obtain valid results. An analysis of the micrographs, as well as that of the fracture surfaces, was conducted using optical microscope and SEM measurements. The chemical composition of the interface layer was determined by EDX measurements. The applied instruments were a VHX-600K optical microscope from Keyence (Osaka, Japan) and a Phenom XL G2 SEM with integrated EDX from ThermoFisher (Waltham, MA, USA).

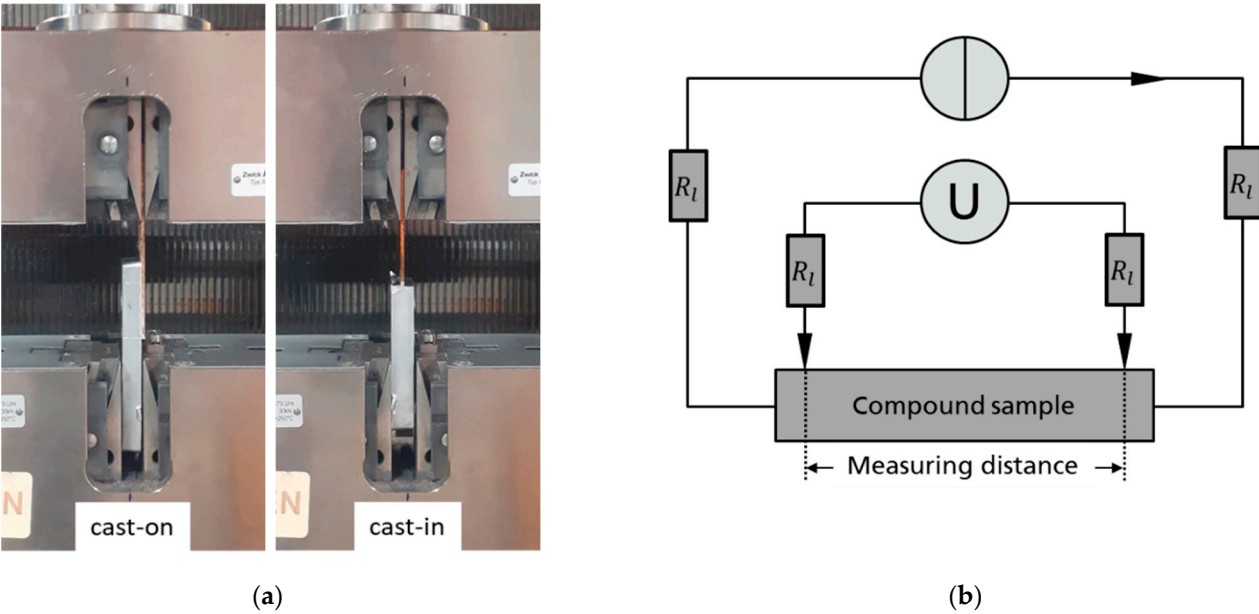

**Figure 3.** Investigation methods used to determine the (**a**) mechanical strength by means of tensile shear testing and (**b**) electrical conductivity of the compound casting samples via the four-terminal sensing method (schematic illustration).

Electrical conductivity was analyzed using the four-terminal sensing method. As schematically depicted in Figure 3b, the voltage was measured before and after the joined area of the sample while an electric current of 10 A was applied. Using Ohm's law, the compound resistance can be calculated from the quotient of the voltage difference and the electric current. The electrical conductivity results from the multiplication of the compound resistance with the quotient of the cross-section and length of the conductor.

To ensure reproducible measurements, a test setup was developed specifically for the given sample geometries (see Figure 2). A sample holder was used to ensure repeatable and precise positioning of the compound samples. Electrical contact of the sample electrodes, each consisting of six contact pins, were attached such that the pins covered the whole width of 40 mm of the sample. Both the contacts and the test probe were spring-loaded, which ensured a constant contact pressure and repeatable positioning. The work was carried out in a temperature-controlled room at 20 °C in order to obtain constant measurement conditions.

## 3. Results and Discussion

### 3.1. Mechanical Strength

In some cases, the mechanical strength of the cast-on specimens was so low that they failed as soon as the castings were removed from the casting machine. Therefore, a higher number of cast-on than cast-in specimens were produced. The mechanical strength values for both specimen geometries were determined by tensile shear testing for $n = 5$ specimens each and are shown in Table 2. The mechanical strength of 2.66 ± 1.05 MPa is rather low and shows a variation of almost 40%. The high variation as well as the failure during removal of the casting demonstrate the instability of cast-on compound specimens and confirms the results from the literature.

**Table 2.** Results of the mechanical testing of the compound casted specimens.

| Geometry | Tensile Force [kN] | Shear Strength [MPa] |
| --- | --- | --- |
| Cast-on | 1.99 ± 0.8 | 2.66 ± 1.05 |
| Cast-in | 6.03 ± 0.2 | 3.85 ± 0.09 |

However, it is surprising that the cast-in specimens show only slightly higher strengths at $3.85 \pm 0.09$ MPa. The insulating effect of the surrounding melt is assumed to result in a longer-lasting temperature impact for the cast-in compared to the cast-on samples. In combination with the approximately doubled contact area, this should result in significantly higher strength values from an increased intermetallic phase formation. Since the aluminum shrinks during solidification, an additional frictional connection between the insert and the shrunk-on aluminum should have occurred. It is supposed that the friction should also have a strength-increasing effect.

Nonetheless, the advantages of the cast-in geometry were observed. Since the shear strength results from the maximum force in relation to the joining area, the values of the cast-in specimens are similarly low compared to those of the cast-on specimens. However, they reach a maximum test load approx. three times higher than the cast-on ones. The variation in the measured values is also significantly lower for cast-in specimens, which indicates a stable process with better reproducibility.

In Figure 4a (showing the fracture surfaces), the cast-in specimens are depicted from each side. The aluminum joining partner was carefully sawn open without damaging the joining areas. The cast-in copper inserts show a necking and elongation of 10%, which is due to inelastic deformation of the material during the test as a result of the comparably high tensile force. The cast-on copper sheets in Figure 4b do not show any change in shape, which is due to the failure of the compound within the elastic deformation limit.

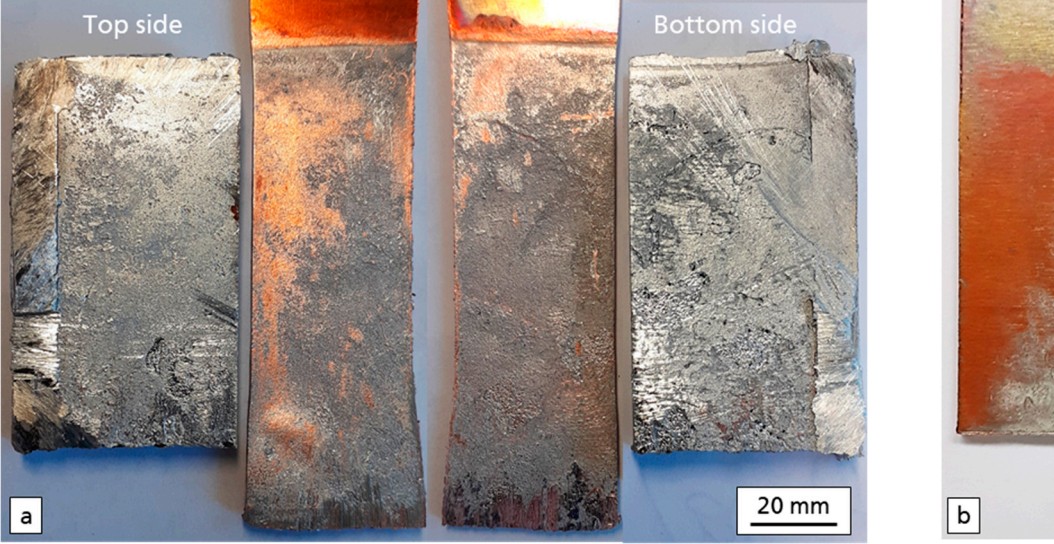 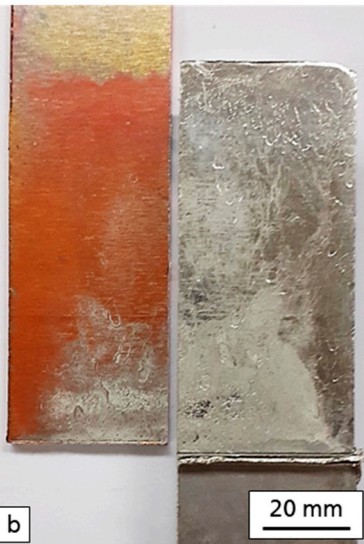

**Figure 4.** Tensile shear tested cast-in (**a**) and cast-on (**b**) compound casting specimens with visible interaction zones on the fracture surfaces.

The analysis of the fracture surfaces in Figure 4 also indicates significant differences resulting from the applied geometries. Only a small amount of grayish discolouration is observed on the cast-on copper insert, which indicates an interaction with the melt. The discolouration occurs only locally in the area of the sprue. In contrast, both sides of the cast-in copper insert in Figure 4 exhibit a significantly larger surface area with visible discolorations. Furthermore, when the sides of the cast-in copper insert are compared, it is revealed that the bottom side has a larger area of discoloration and, thus, more interactions than the top side.

The formation of the interaction areas can be explained by different thermal conditions of the respective cast geometries during the casting process, as well as the subsequent cooling. During mold filling, the molten aluminum flows within milliseconds from the sprue to the end of the joining area of the copper insert. Because the mold temperature of 200 °C is significantly below the melt temperature, part of the heat is dissipated along

this length of the inserts. This results in an inhomogeneous temperature profile, where the thermal impact decreases with increasing specimen length.

For the cast-on geometry, only one side of the copper insert comes into contact with the melt. A large part of the heat is dissipated to the other sides of the insert due to the high thermal conductivity of copper—first through the die mold, and second to the environment. This rapid heat dissipation has a negative effect on the formation of the material bond in the interfacial layer, as the diffusion rate depends strongly on the applied temperature [10]. Therefore, a material bond is formed only in the sprue area of the inserts as visible in Figure 4b. The upper part of the copper insert does not show any changes, which suggests a high rate of cooling. Taking into account the relatively low amount of material bond of the cast-on samples, the maximum tensile force in Table 2 is explained. Thus, the experimental results are identical with those of Freytag.

It can be assumed that the approximately three times higher average maximum test load for the cast-in specimens is due to the significantly enhanced thermal conditions during die casting of the cast-in compared with the cast-on specimens. The heat impact of cast-in specimens occurs on five sides instead of one. Furthermore, the surrounding aluminum has an insulating effect. This reduces the cooling rate and, thus, diffusion is elongated over a longer period of time so that a stronger material bond can be formed. From the literature, shear strengths of 40 MPa to 80 MPa are known from other casting processes such as lost foam [2], horizontal core-filling continuous casting [11], gravity die casting [12], liquid–solid compound casting process [13] and core-filling continuous casting [14]. Compared to the determined shear strength of the cast-in samples, there is a strong difference. This can be explained by the significantly higher thermal impact due to longer casting times, as well as temperatures and higher component thickness. For further analysis, SEM images and EDX measurements were performed and evaluated.

### 3.2. Microstructure of the Interfacial Layer

The aluminum–copper interface layer was examined with micrographs of the compound specimens. The SEM analysis shows an inhomogeneous microstructured interfacial layer with a small amount of material bonds for the cast-on as well as the cast-in samples. Three different structure types can be identified, which will be presented and explained in their formation further below:

I.   Intermetallic phase;
II.  Oxide layer;
III. Defects.

The first two structure types can be seen in the SEM image in Figure 5a of a cast-in sample. Their formation is related to the turbulent mold filling typical of HPDC. When the melt flows over the copper surface, it heats up and shear forces occur. As a result, the oxide layer can partly be ripped off and flushed away from the molten surface of the copper. Thus, pure copper and aluminum come into direct contact and the metals mix to form structure type I—intermetallic phases. Structure type II occurs at the areas where the oxide layer remains or the shear forces are weaker.

This is observed on the SEM image shown in Figure 5b. The copper–aluminum interface is visible over the whole width of the sample area. It can be assumed that the oxide layer in the central area has been removed, allowing for a material bond between aluminum and copper to be formed. This mechanism was also observed in other casting technologies, which apply significantly less pressure. In their investigations of aluminum–copper compound casting using a sand mold under normal atmospheric conditions, Zare et al. [15] describe a similar process of an interfacial layer formation.

The interfacial layer in Figure 5b was analyzed by EDX method and the chemical compositions at locations 1–4 were determined. The measurement results and the suggested intermetallic phases according to the investigation of Braunovic and Aleksandrov [8] are shown in Table 3.

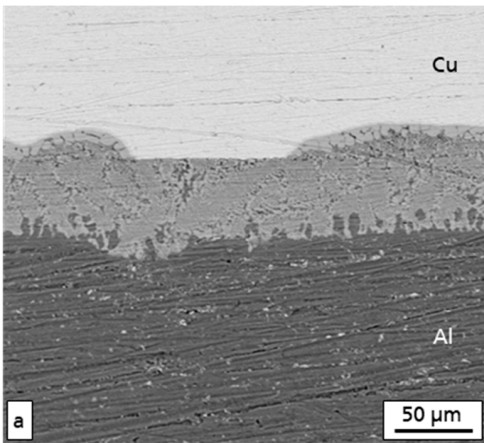 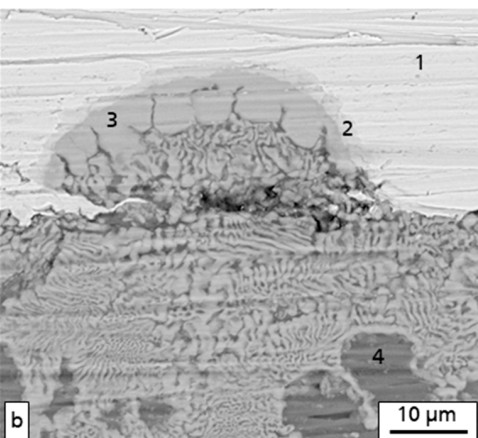

**Figure 5.** SEM image of the interfacial layer of a cast-in specimen with respective material bonding by formation of intermetallic phases as well as separation by oxide layer in (**a**) and detail section of intermetallic phases formation in (**b**).

**Table 3.** Chemical composition at locations 1–4 from Figure 5 and the suggested intermetallic phase according to [8]. The element carbon (C) was disabled for the weight percent calculation.

| Position | Aluminum (wt.%) | Copper (wt.%) | Suggested Phase | Phase Symbol |
|---|---|---|---|---|
| 1 | - | 100.00 | Cu | Cu |
| 2 | 21.3 | 78.7 | $Al_2Cu_3$ | δ |
| 3 | 46.5 | 53.6 | $Al_2Cu$ | θ |
| 4 | 99.2 | 0.9 | Al | Al |

The formation of the material bond by the intermetallic phases causes a change in the shape of the interfacial layer from an initially flat and parallel (left and right side in Figure 5b) to a wavy contour (center part in Figure 5b). The thickness of the δ-phase is 1.5–2 μm (Position 2). It is followed by the θ-phase, which varies in its formation shape and thickness. It changes in the direction of the aluminum from an initially roundish to a lamellar microstructure. The supersaturated α-solid solutions (4 in Figure 5b and Table 3) and lamellar θ-phase build a layer of a fine-structured eutectic. It grows along the interface as shown in Figure 5a, so that it also occurs in areas where a material bond is prevented. This is also described in the investigations of Hu et al. [13]. A continuous eutectic layer on the aluminum side can be observed both for separation by oxide layers and by gaps. The separation by oxide layers is the second type of formation that has been identified. It results from the oxides that was resistant to the shear stresses during mold filling. The contact is plane and takes place without any interaction in the form of diffusion or mixing. Therefore, unlike in the case of intermetallic phase formation, the interface keeps its original shape.

The third type of interface formation are defects, an exemplary case of which is shown in Figure 6. They occur as cracks as in Figure 6a and as cavities in Figure 6b, whereby their formation is attributed to thermal effects during solidification of the aluminum mold.

As previously mentioned, the heat from the molten aluminum causes the copper insert to warm up, causing it to expand. As soon as the temperature balance is reached, the aluminum–copper compound starts to cool down. The change in temperature is accompanied by shrinkage of the metals. Due to the different coefficients of thermal expansion of aluminum with $\alpha_a = 23.8 \times 10^{-6}$ (1/°C) and copper with $\alpha_c = 16.8 \times 10^{-6}$ (1/°C), different length changes occur [10]. This results in shear forces in the interface during cooling. The residual stresses partly exceed the strength of the material bonds, especially when they are as weakly formed as shown in Figure 6a, and cause fracture along the brittle intermetallic phases. In addition to the cracks, long cavities like in Figure 6b are detected in the interfacial layer. However, as can be seen in the micrographs in Figure 7 these occur, unlike the cracks,

only in the cast-in samples. Furthermore, the distribution of shrinking holes and pores can be used to identify where the molten aluminum solidified last. For the cast-on specimen, this is in the center of the aluminum component. As previously mentioned, the cast-on copper insert dissipates a large part of the heat via its surface, so that rapid solidification can be assumed across the joining area. The solidification runs in the direction of the heat center, towards the center of the specimen. So volume deficits caused by solidification and cooling of the aluminum can be compensated by feeding with liquid aluminum and shrinkage holes only occur in the center of the specimen.

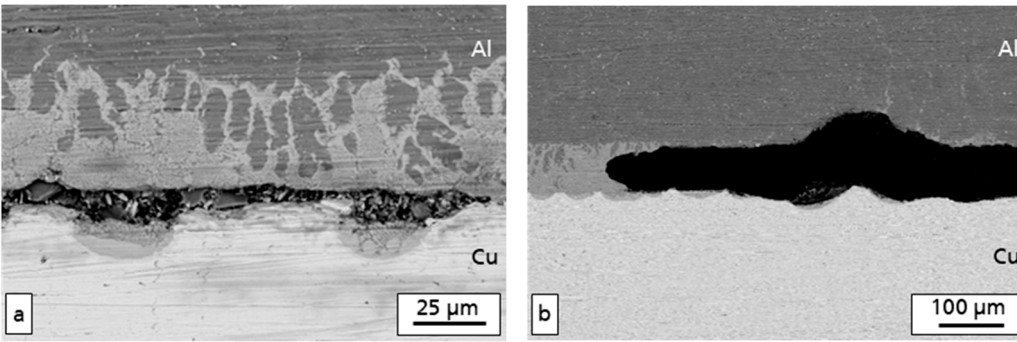

**Figure 6.** Formation of defects in the interfacial layer as cracks (**a**) and cavities (**b**).

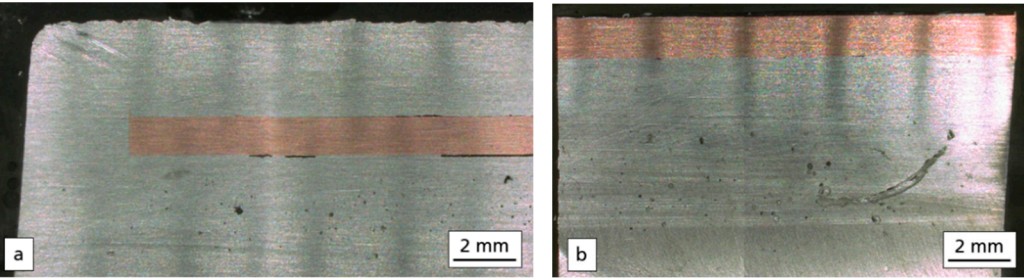

**Figure 7.** Micrographs of the compound specimens of (**a**) cast-in specimen and (**b**) cast-on specimen.

The solidification of the cast-in samples also starts at the copper surface and proceeds in the direction of the outer contour. Because the upper aluminum section is thinner than the lower one, it can be assumed that it cools and solidifies faster. As a result, the compound strength to the upper section is higher than to the lower. When the cooling-induced shrinkage starts, the copper insert moves with the upper aluminum section. The aluminum has already partially solidified due to the previous contact with the lower copper side, so that it can no longer be fed with melt from the lower part of the sample. This results in the formation of cavities or the fracture of already formed compounds. The results are visible in Figure 6. The contour of the copper surface is identical to that of the solidified aluminum. In addition, intermetallic phases have formed on the bottom of the cavity, indicating prior contact with the copper surface. The capillary shape of the interfacial layer also suggests this behavior.

For casting, the general aim is to achieve directional solidification to the center of a component, as in the case of cast-on samples. As a result, strength-reducing defects such as pores and shrinkage cavities do not form in the interfacial layer but in the inner part of the specimen. However, the high cooling rate causes very low diffusion rates, so that a sufficiently strong material bond cannot be achieved. If the sample geometry is changed to enhance the thermal conditions, as in the case of the cast-in-place samples, the number of material bonds increases. Simultaneously, the previously described defects occur close to the interface layer, limiting mechanical strength.

Hu et al. also described the formation of cavities in their investigations as a result of material shrinkage [13]. However, cavities are reported only as a characteristic of a

poor bond formation in the interfacial layer, which can be reduced by increasing the heat impact [2,3,5,16]. By increasing the mold or casting temperature, the thermal impact could be further increased, so solidification is directed to the center of the sample. This would increase diffusion in the interfacial layer and thus the growth of intermetallic phases. However, some studies show that excessively high growth in the intermetallic phase thickness, in turn, leads to a reduction in compound strength. For example, Hu et al. showed the influence of casting time and temperature on the formation of the intermetallic phase thickness. They achieved the highest compound strengths for samples which exhibited the smallest intermetallic phase formation due to low thermal impact [13]. Similar results could be observed for other joining processes. In studies on the growth of intermetallic phases in joined aluminum–copper compounds by friction welding, it was shown that the compound strength decreases linearly as a function of the phase thickness [17]. Abbasi et al. [6] further describe a critical phase thickness of 2.5 μm in their investigations for the cold roll welding process. Above this critical phase thickness, the compound strength decreases drastically and the fatigue behavior changes from elastic to brittle.

Therefore, the thermal conditions in the HPDC process should be adjusted by geometry and process parameters to ensure that solidification can take place in such a way that continuous growth along the interfacial layer can occur. This could be achieved, for example, by preheating the inserts or increasing the pouring or die temperature. The thermal impact has to be just high enough to ensure defect-free interfacial layer formation while minimizing the thickness of intermetallic phases.

### 3.3. Electrical Conductivity

Table 4 shows the results of the electrical resistivity and conductivity measurements of the compound specimens as well as the respective base materials. Theoretically, resistance values between those of aluminum and copper would be possible for an ideal material bond.

**Table 4.** Electrical properties of compound specimens and respective base materials.

| Material | Resistivity [$\Omega \times mm^2/m$] | Conductivity [S/m] |
| :---: | :---: | :---: |
| Cu-ETP | 0.017 | $5.882 \times 10^7$ |
| Al99.5 | 0.029 | $3.448 \times 10^7$ |
| Al-Cu (cast-on) | 0.050 | $2.005 \times 10^7$ |
| Al-Cu (cast-in) | 0.039 | $2.564 \times 10^7$ |

Since the compound samples have higher electrical resistance than the base materials themselves, it can be assumed that losses of resistances are in effect. These occur when the electrical current can only partially pass the contact area. For electrical contacts, the reasons can be the constriction and surface contamination resistance. The constriction resistance occurs, for example, when the contact surfaces of aluminum and copper have a high surface roughness and contact is only realized via roughness peaks. Then, these become "a-spots" and form a smaller contact area compared to the theoretical maximum joining surface, which results in an increase in compound resistance compared to an ideal smooth surface. This is further increased if there are contaminants such as residues of release agents or impurity films due to oxidation or corrosion [18].

The results from the previous sections show the formation of three different types of "structures" within the interfacial layer and that the latter have an influence on the contact area in exactly this way. The locally remaining oxide film and the occurring intermetallic phases form impurity films. Due to their complex crystal structure, they cause a decrease in electrical conductivity [19]. In addition to the higher electrical resistance, particularly the intermetallic phases of aluminum and copper exhibit brittle material behavior, which brings the risk of compound fracture. This, just like the cracks and cavities mentioned above, represents a reduction in the contact area and, thus, an increase in the compound resistance.

In order to evaluate the experimental results in terms of industrial application, the quality factor was used. It describes the ratio of the compound resistance to the conductor resistance of the same length. Since both the conductor cross-section and the conductor material of the joining partners of the compound differ, an average value of resistance is calculated, acting as the reference resistance for the determination of the quality factor. An electrical compound is considered technically suitable if the quality factor ranges between the theoretical possible values of 0.5 and 1 [20]. For the sample geometries applied within this work, this results in a quality factor of

- 0.72 for cast-on samples;
- 0.59 for cast-in samples.

Even if the electrical conductivity of the compound cast geometries is lower than that of the applied base materials, the calculated quality factors for both geometries are below 1. Therefore, they can be classified as technically sufficiently good [20].

Similar investigations on compound casting processes show electrical conductivities of up to $5.29 \times 10^5$ (S/cm) [13]. The electrical conductivities determined in this investigation are significantly lower in comparison. It can be assumed that the electrical conductivity is significantly reduced in particular by the long cavities in the interfacial layer of the compound specimens. If the material bond is enhanced by increasing the temperature impact at the interface during the casting process, the contact area would increase. Accordingly, the electrical resistance would be reduced. With respect to the size of the interfacial defects, the electrical conductivity should, therefore, be further increasable. However, the work of Hu et al. [13] also pointed out a decrease in electrical conductivity in relation to the growth of the intermetallic phases. They report an increase in phase thickness from 25 μm to 300 μm meant a loss in electrical conductivity from an initial $5.29 \times 10^5$ S/cm to $3.83 \times 10^5$ S/cm. The investigations of Braunovic and Aleksandrov [8] confirmed these results. They found linear behavior between the decrease in electrical conductivity with an increase in intermetallic phase thickness and attributed the loss of electrical conductivity to an increase in dislocations and crystal defects. These could be vacancies and interstitial atoms and affect the vibrational capability. The conditions for increasing the electrical conductivity are, therefore, similar to those for increasing the shear strength.

## 4. Conclusions

The investigations showed that the thermal impact for cast-on aluminum–copper specimens is quite low and, therefore, the formation of a material bond is limited. A material bond occurs only locally at positions where the oxide layer on the copper inserts is broken by the aluminum melt during the casting process. If the sample geometry is modified so that the copper sheets are cast-in instead of cast-on, the thermal impact in the interface region can be enhanced and thus the compound strength increased. However, the change in specimen geometry results in a change in setting behavior. This results in the formation of cavities along the interfacial layer. In order to increase the strength and electrical conductivity of the compound, the solidification needs to be directed to the inside of the aluminum sample part to enable homogenous feeding along the entire interfacial layer. However, sufficiently good electrical conductivities can already be achieved for both the cast-on and cast-in specimen geometries. HPDC is, therefore, attractive alternative joining technology for the resource-efficient production of electrically conductive aluminum–copper compounds, in particular for industrial production.

**Author Contributions:** Conceptualization, N.N. and T.L.; methodology, N.N.; validation, N.N.; formal analysis, N.N.; investigation, N.N.; resources, T.L. and B.M.; data curation, N.N.; writing—original draft preparation, N.N.; writing—review and editing, T.L. and B.M.; visualization, N.N.; supervision, T.L.; project administration, N.N., T.L. and B.M.; funding acquisition, T.L. and B.M. All authors have read and agreed to the published version of the manuscript.

**Funding:** This research received no external funding.

**Institutional Review Board Statement:** Not applicable.

**Informed Consent Statement:** Not applicable.

**Data Availability Statement:** Not applicable.

**Conflicts of Interest:** The authors declare no conflict of interest.

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
