# Peer review of "Investigation of the Microstructure and Properties of Aluminum–Copper Compounds Fabricated by the High-Pressure Die Casting Process"

_metals, doi:10.3390/met12081314_

Round 1

Reviewer 1 Report

Nane Nolte et al., elaborated the manuscript titled ”Investigation of the Microstructure and Properties of Aluminum-Copper Compounds Fabricated by the High-Pressure Die Casting Process”, addressing the casting process following aluminum and copper, with high economic and technical impact in the electrical contact industry. Elaborated studies were in detail presented and oriented in interfacial analysis by SEM and EDX measurements for the aluminum-copper compounds followed by mechanical and electrical investigations. Before acceptance, authors should consider a minor revision of the manuscript and implement the following observations:

-          Between keywords, authors used HPDC; This abbreviation should be used in the entire manuscript;

-          Introduce missing bracket “Table 40. mm x 100 mm x 1.5 mm)”;

-          Figure 2 could include XYZ coordinates for better visibility and understanding;

-          Relating to the results presented in table 2, why the shear strength error is different between cast in and cast on results? If is related to the sample geometry, than it should be stated;

-          As an suggestion, section “materials and methods” could include the mechanical experimental configurations (example: the positioning of the sample during the tensile shear test etc.); Could be of highly added value to this section that is too summarized;

-          The SEM measurements (fig. 4 and 5) addressing the interfacial layer were conducted in cutted section of the molted samples. However, this results from observing fig. 6 and not directly from the text. How were the sample prepared for these measurements etc. should be briefly mentioned;

-          In figure 6, avoid using thousands of microns in the scale bar;

-          As an observation, investigating defects by SEM measurements on selected surfaces is more dependent on individuals luck, or relies on cutting multiple sections to be certain of the outcomes (presence of pores, cracks, elongated or spherical voids etc.); Authors could consider applying a NDT as a XCT method to assist the SEM findings and in order to better trace the presence of defects at the interface;

-          Table 2 shows the results of the electrical resistivity and conductivity”; I think the authors are referring to Table 4;

Author Response

Thank you very much for the feedback and helpful tips as well as for your time! I have added all the changes. The study of the interfacial layer was examined at several points by means of micrographs in order to obtain valid results. In fact, I am currently testing the samples using CT analysis because I am researching strength enhancing treatments. 

Reviewer 2 Report

This paper shows some useful experimental results. Questions and Comments are listed below.

1.     Figure 1. Is there any difference in cast-in and cast-on process in the joining process shown in Figure 1? Please clarify this point.

2.     Figure 2. Please indicate Cu and Al on the figure. It would be easier to read.

3.     Figure 5. This seems to be the drawback of this process. Any improvement suggestion for future work?

Author Response

Thank you very much for the feedback and for your time! 

1: The mold includes two cavities each for cast on and cast in. All four inserts are positioned and then wetted with melt, two on one side ( cast-on) and two on five sides ( cast-in). 

This is explained above Fig. 2 "The applied casting mold allowed for the creation of two different compound geome-tries, so that the copper inserts can be cast-on to an area of 40 mm x 40 mm (Figure 2a) as well as cast-in 40 mm deep (Figure 2b)."

2: I have added the designations
3: I have added possible improvements to the text. 
My next publication will discuss exactly this problem. Different casting parameters as well as surface treatments will be investigated